# Genetic Loss of miR-205 Causes Increased Mammary Gland Development

**DOI:** 10.3390/ncrna10010004

**Published:** 2023-12-31

**Authors:** Alessandra Cataldo, Douglas G. Cheung, John P. Hagan, Matteo Fassan, Sukhinder Sandhu-Deol, Carlo M. Croce, Gianpiero Di Leva, Marilena V. Iorio

**Affiliations:** 1Research Department, Fondazione IRCCS Istituto Nazionale dei Tumori, 20133 Milan, Italy; 2Comprehensive Cancer Center, Department of Cancer Biology and Genetics, The Ohio State University, Columbus, OH 43210, USA; 3Department of Neurosurgery, The University of Texas Health Science Center at Houston, Houston, TX 77054, USA; 4Department of Medicine, DIMED, University of Padua, 35122 Padua, Italy; 5Veneto Institute of Oncology, IOV-IRCSS, 35128 Padua, Italy; 6School of Pharmacy and Bioengineering, Keele University, Keele ST5 5BG, UK

**Keywords:** microRNA, mammary gland, breast cancer

## Abstract

MiRNAs play crucial roles in a broad spectrum of biological processes, both physiological and pathological. Different reports implicate miR-205 in the control of breast stem cell properties. Differential miR-205 expression has been observed in different stages of mammary gland development and maturation. However, a functional role in this process has not been clearly demonstrated. We generated an miR-205 knockout in the FVB/N mouse strain, which is viable and characterized by enhanced mammary gland development. Indeed, mammary glands of miR-205^−/−^ female mice at different ages (1.5 and 5.5 months) show increased outgrowth and branching. This evidence is consistent with our previously reported data demonstrating the direct miR-205-mediated targeting of HER3, a master regulator of mammary gland development, and the oncosuppressive activity of this microRNA in different types of breast cancer.

## 1. Introduction

MiRNAs have rapidly emerged as a major class of regulatory genes controlling development and disease processes in mammals [1]. Despite the increasing number of correlation studies, few engineered mouse models have demonstrated a causal link between a specific miRNA and physiological functions.

Previous studies have reported that genetic loss of miR-205 causes perinatal lethality [2] due to an altered PI(3)K pathway in the epidermis, essential to maintain stem cell self-renewal [3], even though the effect might be strain-specific.

The role of miR-205 has been widely described in breast cancer. Indeed, our group and others have demonstrated that this miRNA is downregulated in tumors [4] and exerts an oncosuppressive role by targeting HER3 [5,6], E2F1 and LAMC1 [7], ZEB1, and ZEB2 [8], VEGF [6], NOTCH2 [9], with consequent effects on cell proliferation, migration, stem cell properties, and response to therapy.

MiR-205 has also been associated with the physiological development of the mammary gland, even though a direct role has not been clearly demonstrated yet. MiR-205 is strongly expressed in the basal epithelium until the mature virgin stage, whereas it is absent in the luminal compartment [10,11]. In addition, expression of miR-205 increases in both basal and luminal epithelium during pregnancy and lactation, with a subsequent increase in expression during late involution. Cell culture models of normal mammary epithelial cells showed that miR-205 is highly expressed in the “progenitor-like” subpopulation cells [12], suggesting a role for miR-205 in mouse mammary gland stem cells. The same authors [13] confirmed the role of miR-205 in the regenerative potential of stem cells by performing transplantation of WT and miR-205^−/−^ mammary epithelial cells (MECs), even though differences in the extent of fat pad filling were significant only under stress conditions. In contrast, a report by Chao and colleagues [9] suggested that the loss of miR-205, induced by Jagged1, is associated with increased stemness properties in human breast cancer. These authors also showed that miR-205 knockdown, obtained by intramammary infection of a lentivirus encoding for a miRNA-shRNA, caused increased ductal length and hyperplastic lesions.

Here, we generated a conditional miR-205 knockout mouse to study the physiological and developmental role of miR-205 in mice and observed that genetic loss of this miRNA caused increased mammary gland outgrowth and branching.

## 2. Results and Discussion

### 2.1. Generation of miR-205KO Model

We generated a conditional miR-205 knockout FVB/N mouse, taking advantage of the well-described Cre-loxP technology. The structures of the wild-type allele, the targeting vector, and the targeted allele are reported in the cartoon in Appendix A. To generate an obligatory knockout that was used in this study, we crossed our miR-205 conditional mice with the ubiquitously expressed EIIa-cre mouse and screened the genotype using the Southern Blot method (Appendix A). Since we generated a ubiquitous KO model, Southern Blot was also performed in different tissues (Appendix A), and the loss of expression was assessed by Northern Blot (Figure 1A) and qRT-PCR methods (Figure 1B).

In the mammary gland, miR-205 was also assessed by ISH, which revealed a preferential localization of this microRNA in the mioepithelial compartment (Figure 2) consistent with previous reports [11] and confirmed the loss of the expression in KO mice.

### 2.2. MiR-205KO Leads to Increased Mammary Gland Development

Previously, Wang and colleagues reported that miR-205 knockout leads to neonatal lethality (10 days after birth) in C57BL/6 mouse strain due to skin defects [3].

Instead, our miR-205 knockout mice on an FVB/N genetic background are viable, suggesting that there are strain-specific modifiers that alter the functional consequences of miR-205 deletion. Our knockout mice lacked overt skin phenotypes. There was no statistically significant difference in mouse numbers relative to Mendelian ratios, although fewer miR-205^−/−^ were observed from heterozygous crosses (miR-205^+/+^, ^+/−^, and ^−/−^ mice were n = 22 (25.6%), 48 (55.8%), and 16 (18.6%), respectively).

Considering the evidence reported in the literature of an increased expression of miR-205 in ScaI+ cells [12] and thus its potential involvement in the self-renewal of breast stem cells, we reasoned that genetic loss of miR-205 could affect mammary gland development and function. To test this hypothesis, we performed pup growth curve analysis. Specifically, we measured pup weight every day for 21 days after the birth date derived from crosses of wild-type males with either WT or KO females. To our surprise, miR-205 null female mice were perfectly able to feed their pups.Consistent with this observation, whole-mount analysis of miR-205^+/+^ and ^−/−^ mammary glands at different stages of mammary gland development did not show defects in their structure and development.

Unexpectedly, whole-mount analysis of mammary glands collected at different ages, during puberty at 6 weeks of age and in mature virgins at 5.5 months, show that branching is increased in KO mice (even though statistically significant only at 6 weeks of age, *p* = 0.00016) (Figure 3), and outgrowth was more prominent (Figure 4; see the black line in 1× panels and the table, statistically significant in mice both 6 weeks of age, *p* = 0.0025, and in 5.5-month-old mice, *p* = 0.0018).

In addition, a qualitative evaluation of the H&E staining of mammary glands from mice 6 weeks of age showed that, despite similar histology between WT and KO, terminal end buds are more prominent and are also distributed in the central part of the gland (Figure 5A). On the other hand, normal ducts in KO are rare. Similarly, analyses performed on 5-month-old female mice (Figure 5B) showed that miR-205^−/−^ ducts had pseudostratified and hyperplastic epithelia with frequent mitoses (arrow) (right panels).

These data are consistent with our previous report demonstrating the direct miR-205-mediated targeting of HER3, a master regulator of mammary gland development, and the oncosuppressive activity of this microRNA in different types of breast cancer.

For this reason, we also investigated the potential role of miR-205 in the development of HER2+ breast tumors. Interestingly, preliminary data indicate that miR-205 expression is lowered in mammary tumors generated in MMTV-neu transgenic mice (Appendix A), and crossing miR-205^−/−^ mice with mice carrying heterozygous neu increases the number and volume of lesions and the number of lung metastases.

This is not surprising if we consider the functional role attributed to miR-205 in the breast cancer model, where miR-205 plays an oncosuppressive role, controlling cell proliferation and survival by regulating the expression of several targets [14].

## 3. Discussion

The involvement of microRNAs in physiological and pathological processes is currently well recognized. In particular, evidence underlines how not only is microRNA expression differently regulated in different phases of mammary gland development, but also plays a functional role.

Since functional studies available to date on miR-205 function mainly rely on ectopic overexpression or silencing of miR-205 in cell culture models, the generation of loss-of-function mouse models for miR-205 represented an opportunity to clarify the role of this miRNA in normal development and cancer.

Our data demonstrate, using a Cre-Lox KO model, that miR-205 loss causes improved mammary gland development. Inconsistent with a previous study by Wang and colleagues [3], our miR-205 knockout was viable and did not show any skin defects. This discrepancy might be due to the different mouse strains used (FVB/N genetic background versus C57BL/6). Considering the reported role of miR-205 in normal mammary stem cells [12,13], we expected that genetic loss of this miRNA would cause defects in mammary gland development, impaired milk production, and capability to feed the pups. Instead, we observed increased mammary gland outgrowth and branching of terminal end buds in female miR-205 KO mice.

Of note, one of the potential targets of miR-205 is HER3, as we [5] and others have previously demonstrated, which is not only the preferential partner of HER2 but is also required for ductal morphogenesis in the mouse mammary gland [15].

Considering the role of HER3 in the development and progression of HER2+ breast cancer and our preliminary data suggesting that loss of miR-205 might promote HER2-driven tumorigenesis, it is not surprising that low expression of this miRNA is a feature of HER2+ breast cancer, as also reported in series of human tumors [16].

Unfortunately, our mammary-specific KO mouse was not suitable since, after crossing with MMTV-Cre, the deletion of the target was not efficient; however, our model holds the potential to study the tissue-specific functional role of miR-205.

In conclusion, genetic deletion of miR-205 in the FVB mouse strain is viable and impacts mammary gland development by increasing outgrowth and branching.

## 4. Materials and Methods

Generation of miR-205KO model. The structures of the wild-type allele, the targeting vector, and the expected targeted allele are reported in Appendix A. The pGK-Neomycin locus (neo) and thymidine kinase (TK) represent the positive and negative selection markers, respectively. Yellow ovals delimiting the ‘neo’ cassette are FRT sequences for the removal of the ‘neo’ cassette in vivo. The miR-205 gene (blue) is flanked by 2 loxP sequences in order to obtain its deletion in vivo by crossing the obtained mouse strain carrying miR-205 conditional alleles (miR-205Co/Co) with EIIA-Cre transgenic mice. Location of 5′ and 3’ hybridization probes and BclII sites used in Southern blot analyses are shown. Sizes of expected DNA fragments for these analyses are depicted for the corresponding genomic regions (not drawn to scale).

After the transfection of the targeting vector into ES cells and selection for clones positive for the homologous recombination, targeted ES cell clones were used to generate mice heterozygous for the miR-205 targeted allele. These mice were then crossed to EIIA-Cre expressing mice to remove the loxP-flanked miR-205 cassette and generate miR-205^+/−^ mice. After backcrossing (n = 6) to wt FVB, heterozygous mice were then crossed to obtain miR-205^−/−^ mice, and littermates were always used as controls.

Southern Blot. For mouse genotyping, genomic DNA was isolated from tail and digested with BclII, and Southern Blot method was performed using two different probes (5′ and 3′ probes), as previously reported. Briefly, 7 ug DNA was digested with BclII, run on 0.7% agarose gel, and transferred to Hybond-N+ membranes (Amersham), then incubated with radioactive probes (p-ATP):

The 5′ probe was amplified with the following primers:Fw: 5′-CACTCCAGGCTTCTGGGTCAAAGAACTAG-3′Rw: 5′-AAGGTTCCTTTGAACTGAATCTGAGAGG-3′

The 3′ probe was amplified with the following primers:Fw: 5′-GAACAACTGTAGTGAGCCCATAGATGAAC-3′Rw: 5′-CAAGCACTGTCTGATTTTCACACCAGCAG-3′

As graphically described in Appendix A, after digestion with BclII enzyme, the designed probes enable the discrimination of the presence of the cko allele, characterized by the presence of a 6.9 kb or 5.5 kb band (with the 5′ or 3′ probe, respectively), in addition to the band corresponding to the wt allele (10.5 kb). After removal of the loxP cassette by crossing with EIIA-Cre mice, the 5′ probe would identify a band of 6 kb corresponding to the KO allele.

Northern Blot for miR-205. MicroRNA Northern Blot analysis was performed as previously described [17] using a miR-205 antisense probe (5′-CAGACTCCGGTGGAATGAAGGA-3′). Briefly, RNA samples (10 μg each) were run on 15% polyacrylamide and 7M urea Criterion precast gels (Bio-Rad) and transferred onto Hybond-N+ membranes (Amersham). Hybridization was performed at 37 °C in ULTRAhyb-Oligo hybrization buffer (Ambion) for 16 h. Membranes were washed at 37 °C, twice with 2× saline-sodium phosphate-EDTA and 0.5% SDS.

Real-Time PCR. TaqMan MicroRNA Reverse Transcription kit and TaqMan MicroRNA Assay were used to detect and quantify mature microRNA-205 in accordance with manufacturer’s instructions (Applied Biosystems). Normalization was performed with murine sno-135 RNA.

Whole-mount analysis of the mammary glands and E&E. Whole-mount carmine-alum staining of mammary glands was performed as previously described [18]. Briefly, inguinal mammary glands were dissected from mice, spread onto glass slides, fixed in a 6:3:1 mixture of ethanol:chloroform:glacial acetic acid, hydrated, stained overnight in 0.2% carmine (Sigma) and 0.5% AlK(SO_4_)_2_, dehydrated in graded solutions of ethanol, cleared in Histoclear, and mounted.

Branching and outgrowth evaluation. Each scanned image of whole-mounted mammary glands was aligned and divided with the same six imaginary lines by positioning their central lymph nodes one-quarter of the way along from the left of the entire mammary gland (Goel HL et al.; Development 2011) [19]. Outgrowth is measured as positioning of the TEBs (terminal end buds), structures that form at the tips of ducts, a measure of how the gland is invading the fat pad. Branching is the sum of number of mammary ducts crossing each line (scheme of quantification in Figure 3A).

Animal studies. All animal studies were performed in accordance with the relevant guidelines and regulations and with the approval of the responsible local and national authorities.

Histological staining. Dissected mammary glands were fixed in 4% paraformaldehyde and then either frozen in cryomatrix or embedded in paraffin. Paraffin sections (7 μm) were used for hematoxylin–eosin staining or in situ hybridization.

In situ hybridization. MiR-205 expression localization was assessed by in situ hybridization, as previously reported [20], using a commercially available LNA probe (Exiqon).

Statistics. Data are expressed as mean ± s.d. Statistical comparisons were tested by Student’s *t*-test using Graphpad Prism 5 software. *p* < 0.05 was considered statistically significant.

## Figures and Tables

**Figure 1 ncrna-10-00004-f001:**
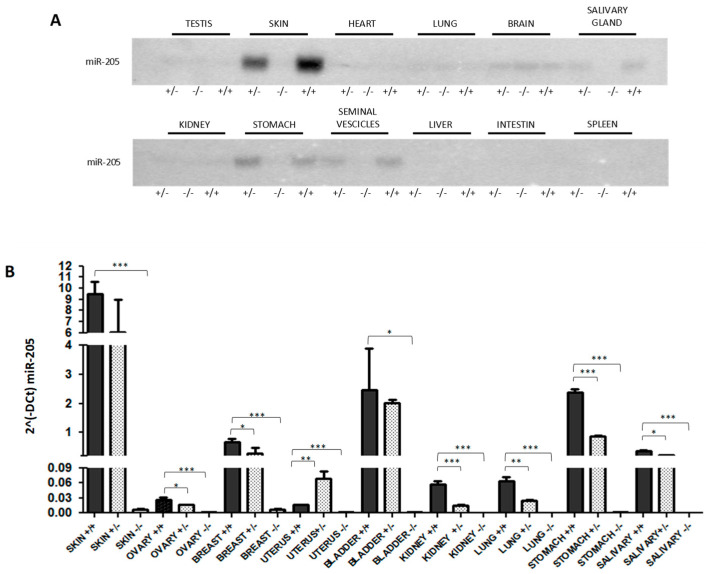
miR-205 expression evaluated by Northern Blot (**A**) and qRT-PCR (**B**) on different organs from miR-205 WT (+/+), KO (−/−) and heterozygous (+/−) mice. Images are representative. Bars indicate SD of three replicates. * *p* < 0.05; ** *p* < 0.01, *** *p* < 0.001.

**Figure 2 ncrna-10-00004-f002:**
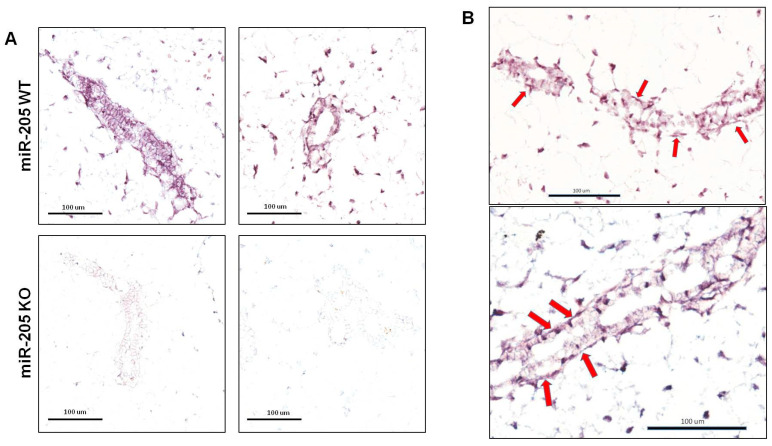
miR-205 expression and localization. In situ hybridization of miR-205 in mammary glands confirmed the loss of expression in KO mice versus WT (**A**); in addition, a more detailed evaluation revealed preferential localization of this miRNA in the mioepithelial cells ((**B**), red arrows). Images are representative.

**Figure 3 ncrna-10-00004-f003:**
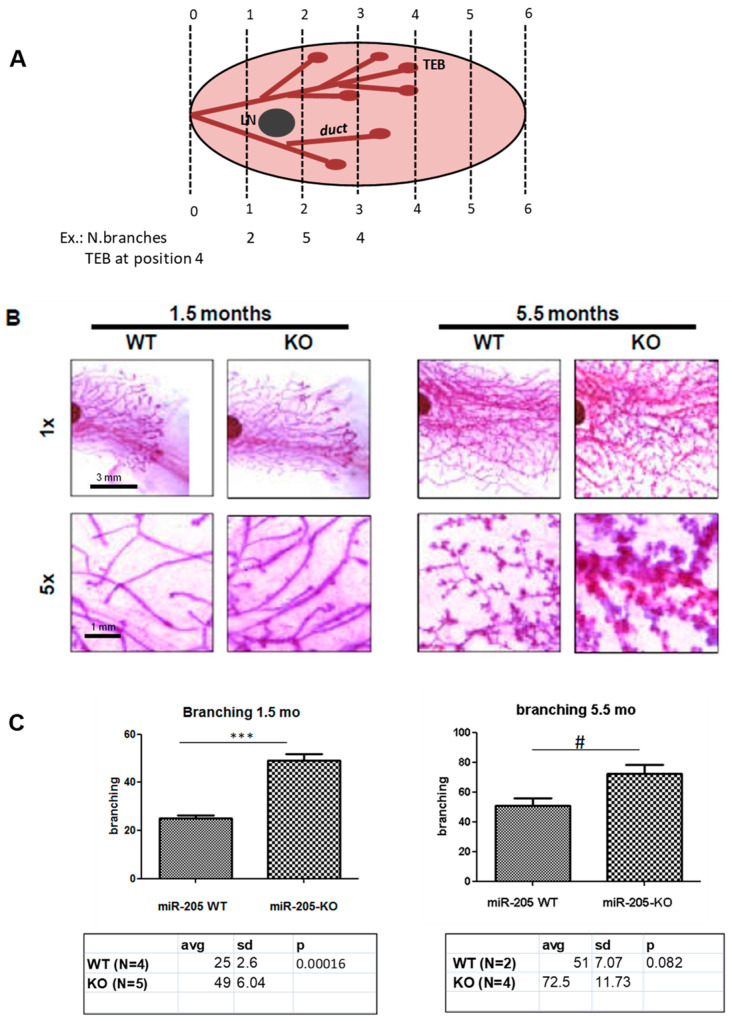
Genetic loss of miR-205 induces increased mammary gland branching. Mammary glands from 6-week (**left** panels) and 5.5-month (**right** panels)-old female mice (WT and miR-205 KO) have been evaluated. Scheme of quantification method, according to the protocol reported by Goel HL et al.; *Development,* 2011 (please see Materials and Methods for details), is shown in the image in (**A**) (LN= lymph node; TEB= terminal end bud). Whole-mount analysis (**B**) shows increased branching in miR-205 KO mice at both ages, even though we reach statistical significance only in 6-week-old mice (quantification in (**C**)). Images are representative. *** *p* < 0.001; # *p* < 0.1 (n.s.)

**Figure 4 ncrna-10-00004-f004:**
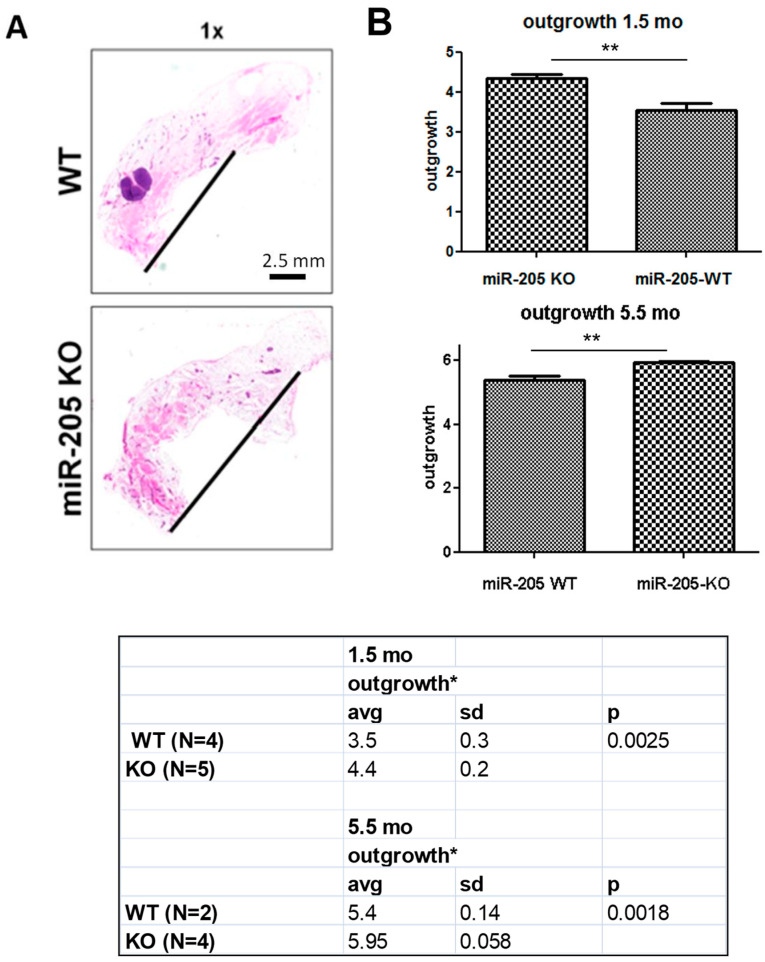
Genetic loss of miR-205 induces increased mammary gland outgrowth. Mammary glands from 6-week- and 5.5-month-old female mice (WT and miR-205 KO) have been evaluated according to the protocol reported by Goel HL et al.; *Development,* 2011 (please see Materials and Methods for details), and pictured in Figure 3A. Histological evaluation of H&E stained glands reveals increased outgrowth in KO mice ((**A**), black line), quantified in (**B**) and reported in the table. Images are representative of mammary gland outgrowth at 6 weeks of age. ** *p* < 0.01, * *p* < 0.05.

**Figure 5 ncrna-10-00004-f005:**
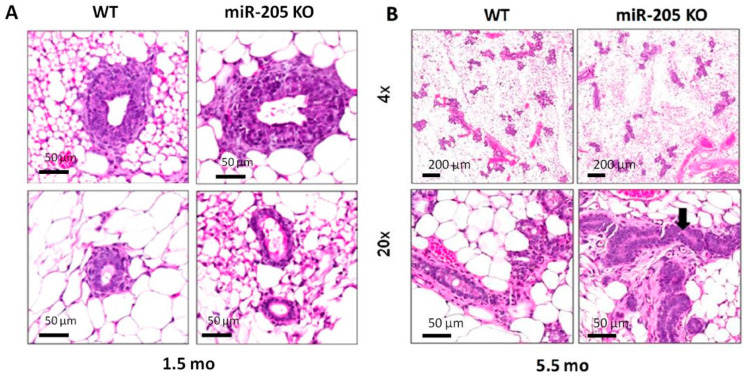
Qualitative evaluation of the H&E staining of mammary glands from 6 weeks of age mice shows that, despite similar histology between WT and KO, terminal end buds are more prominent and distributed in the central part of the gland ((**A**), **left** panels). On the other hand, normal ducts in KO are rare ((**A**), **right** panels). Similarly, analyses performed on 5-month-old female mice (**B**) showed that miR-205^−/−^ ducts had pseudostratified and hyperplastic epithelia with frequent mitoses (arrow) (**right** panels).

## Data Availability

All data are reported in the manuscript and in the Appendix A.

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
