# Peer review of "Genetic Loss of miR-205 Causes Increased Mammary Gland Development"

_ncrna, 2023, doi:10.3390/ncrna10010004_

Round 1
Reviewer 1 Report
Comments and Suggestions for Authors
Thank you for the opportunity to review the manuscript titled “Genetic loss of miR-205 causes mammary gland hyperplasia” by Cataldo et al. In this work the authors find that the deletion of miR-205 leads to increased outgrowth and branching of mammary glands in mice. The novelty of the manuscript is the generation of the first miR-205 conditional knockout mice. However, the authors generate knockouts and characterize the effects in mammary glands. Consistent with previous studies indicating miR-205 functions as an oncosuppressive miRNA in breast cancer (and others), the authors find miR-205 deletion leads to increased outgrowth and branching of mammary glands, but not tumors. Intriguingly, their knockout of miR-205 contrasts with previous publications indicating embryonic lethality. Prior to publication there are some concerns to note.
1. The authors claim in the abstract and title that miR-205 deletion results in hyperplasia and increase proliferation of terminal end buds. However, there is no quantitative data to support this conclusion. The authors should perform Ki-67 immunohistochemistry, count mitotic fields, or conduct a similar analysis to draw this conclusion.
2. Tables for Figure 2 are not appropriate for quantitative data presentation. The authors should present the data in a manner that the error is visualized (bar plot or box plot with the error indicated).
3. The authors should define in the materials and methods the process for quantifying branching and outgrowth, also units should be added to the tables for figure 2 (or better yet to the new box plots). The authors should also describe the number of mice and fields that are quantified.
4. The authors should better describe what outgrowth of the mammary gland is measuring. The histologic difference of outgrowth is not apparently obvious without an explanation for non-mammary gland experts. The histologic analysis in general should be better explained for non-experts. The arrow is not defined in the figure 2 legend, nor is it clear that a single mitosis event is significant (hence the need for quantitative analysis (comment 1). The authors note apoptotic bodies, but do not show histologic evidence of this nor explain its significance.
5. The authors should expand the introduction with discussions of the previously generated miR-205 knockdown/knockout models (references 12 and 13, Lu et al. Stem Cells 2018 doi.org/10.1002/stem.2914, and finally Park et al. Cell Reports 2012 doi.org/10.1016/j.celrep.2012.02.008).
6. It is unclear why a conditional allele was generated and then crossed to a global Cre (EIIa-Cre). Were the animals in the study backcrossed and true knockouts or F1s from (EIIa-Cre;miR-205cHet x miR-205cHet). This should be clarified in the methods and may have implications for the viability of the mice.
7. In general, the figure legends should be written more clearly, especially the supplemental figure legends. Details of the methodology, samples, sample size, and statistical analysis are insufficient. Statistical significance is not indicated (S3) and axes are not labeled. Band sizes are useful, but should also label what the band is indicating on blots in S1 (as is shown in S2).
8. In the discussion, the authors should expand the reasoning and hypotheses on differences between the Wang et al. and the Park et al. lethality phenotype with their viable phenotype. The authors need to reference and describe the Park et al. study referenced above in comment 5.
9. The authors should consider higher resolution images for Figure 1 (zoom in on the glands since most of the image is not relevant.
10. The authors should also consider including much of the data in the supplement in their main figures.
11. Figure S4 description and data are not explained. What skin phenotype is being measured and what does that have to do with the findings.
12. It may not be appropriate to reference unpublished preliminary data in the discussion (lines 127-130). Instead, the authors should consider discussing models in which this cKO will be useful to investigate in future work.
Author Response
We thank the reviewer for the useful comments and suggestions, which certainly helped improving the manuscript.
REVIEWER 1.
- The authors claim in the abstract and title that miR-205 deletion results in hyperplasia and increase proliferation of terminal end buds. However, there is no quantitative data to support this conclusion. The authors should perform Ki-67 immunohistochemistry, count mitotic fields, or conduct a similar analysis to draw this conclusion.
Thank you for this comment. Indeed, we have quantified branching and outgrowth (according to the reference Goel GL et al 2011) in the mammary gland, but we do not have a quantitative analysis of the proliferation status. We only have a qualitative assessment of a more pronounced terminal end bud proliferation in the miR-205 KO mice, with ducts presenting pseudostratified and hyperplastic epithelia, with frequent mitoses. Unfortunately, we do not have material left to assess Ki67. We thus mitigated the assertion in the title and abstract, where we now refer only to the data concerning branching and outgrowth, and we report the qualitative evaluation as description in the text.
Considering branching and outgrowth, we better explained the protocol and quantification in the material and methods section.
- Tables for Figure 2 are not appropriate for quantitative data presentation. The authors should present the data in a manner that the error is visualized (bar plot or box plot with the error indicated).
We modified the figure reporting the complete data in the table, and a boxplot to visualize the data. Please note that previous figure 2AB has been split in two different figures, Figure 3 and 4 in the revised version of the manuscript.
- The authors should define in the materials and methods the process for quantifying branching and outgrowth, also units should be added to the tables for figure 2 (or better yet to the new box plots). The authors should also describe the number of mice and fields that are quantified.
Thank you for this comment. We modified the figures as requested and added the required info. In addition, we better specified the procedure for branching and outgrowth quantification in material and methods section.
- The authors should better describe what outgrowth of the mammary gland is measuring. The histologic difference of outgrowth is not apparently obvious without an explanation for non-mammary gland experts. The histologic analysis in general should be better explained for non-experts. The arrow is not defined in the figure 2 legend, nor is it clear that a single mitosis event is significant (hence the need for quantitative analysis (comment 1). The authors note apoptotic bodies, but do not show histologic evidence of this nor explain its significance.
Considering mammary gland branching and outgrowth evaluation, we specified the protocol and analysis criteria in the material and method section. In addition, we modified the figures and added the missing information both in the table and boxplot. Concerning the histological evaluaation, we better detailed the description both in the text and in the figure legends; in addition we specified that the presence of a more abundant mitotic bodies and proliferative epithelium is only a qualitative observation.
- The authors should expand the introduction with discussions of the previously generated miR-205 knockdown/knockout models (references 12 and 13, Lu et al. Stem Cells 2018 doi.org/10.1002/stem.2914, and finally Park et al. Cell Reports 2012 doi.org/10.1016/j.celrep.2012.02.008). We better discussed the previously generated miR-205 KD/KO models in both introduction and discussion.
- It is unclear why a conditional allele was generated and then crossed to a global Cre (EIIa-Cre). Were the animals in the study backcrossed and true knockouts or F1s from (EIIa-Cre;miR-205cHet x miR-205cHet). This should be clarified in the methods and may have implications for the viability of the mice.
We thank the reviewer for the opportunity to clarify. The animals are indeed true knockouts. We initially generated both ubiquitous (crossed to EIIA-Cre) and mammary-specific (crossed to MMTV-Cre) miR-205 KO mice, however in the latest the removal of the loxP cassette was inefficient, probably due to a poor expression of the Cre enzyme under the MMTV promoter in our model. We thus decided to perform our studies in the ubiquitous KO, which was viable.
- In general, the figure legends should be written more clearly, especially the supplemental figure legends. Details of the methodology, samples, sample size, and statistical analysis are insufficient. Statistical significance is not indicated (S3) and axes are not labeled. Band sizes are useful, but should also label what the band is indicating on blots in S1 (as is shown in S2).
Thank you for this comment. We improved both figures and figure legends adding the statistical analysis where missing and a more detailed description. Concerning Figure S1, we modified the figure and clarified the description in the figure legend.
- In the discussion, the authors should expand the reasoning and hypotheses on differences between the Wang et al. and the Park et al. lethality phenotype with their viable phenotype. The authors need to reference and describe the Park et al. study referenced above in comment 5.
We better discussed the previously generated miR-205 KD/KO models in both introduction and discussion (see point 5).
- The authors should consider higher resolution images for Figure 1 (zoom in on the glands since most of the image is not relevant.). We modified the Figures adding new images (even though at the same resolution) that should be more representative of miR-205 expression and localization.
- The authors should also consider including much of the data in the supplement in their main figures. We thank the reviewer for the suggestion. We indeed moved to the main text most part of the supplementary figures.
- Figure S4 description and data are not explained. What skin phenotype is being measured and what does that have to do with the findings. Thanks for the comment, I do apologize, there was a mistake since Figure S4 was referred to pup growth curve, whereas skin phenotype is actually not reported (data not shown). However, we better explained in the text what we observed and the reason why we actually checked this phenotype, since it was previously reported that KO of miR-205 causes embryonic lethality due to skin defects (even though in a different mouse strain).
- It may not be appropriate to reference unpublished preliminary data in the discussion (lines 127-130). Instead, the authors should consider discussing models in which this cKO will be useful to investigate in future work. Thank you for this note. We moved the comment on preliminary data in the results sections, and improved the discussion underlining the potential of the conditional KO model for future work.

Reviewer 2 Report
Comments and Suggestions for Authors
In this manuscript Caltaldo et al engineered a miR-205 knockout mouse model to understand the role of miR-205 in mammary gland development. Their data suggests that miR-205 loss causes mammary gland hyperplasia. However the data is not sufficient to support the claim demonstrating the direct miR-205-mediated regulation of mammary gland development.
- miR-205 has been shown to play an oncosuppressive role, controlling cell proliferation and survival by regulating the expression of several targets. It would be interesting to check the expression of these target genes in miR-205 KO mammary glands via RNA-seq followed by downstream validation studies.
- Expression of HER2/3 should be evaluated in miR-205 KO mice
Author Response
REVIEWER 2.
- miR-205 has been shown to play an oncosuppressive role, controlling cell proliferation and survival by regulating the expression of several targets. It would be interesting to check the expression of these target genes in miR-205 KO mammary glands via RNA-seq followed by downstream validation studies.
- Expression of HER2/3 should be evaluated in miR-205 KO mice
- We agree with the reviewer that it would be interesting to assess the expression of miR-205 target genes in the mammary gland of the KO mice. Unfortunately we do not have material left available to perform RNA-seq analysis.
- However, as suggested by the reviewer, we made an attempt to analyze HER3 expression by IHC in sections from miR-205 AND KO mice at both 1.5 and 5.5 months of age, unfortunately in our hands the staining was extremely faint, not evaluable. We did not have material left to assess protein expression by Western blot.

Reviewer 3 Report
Comments and Suggestions for Authors
miR-205 is a highly conserved microRNA across species. It is expressed in various types of epithelial tissues, such as mammary gland, and exhibits both oncogenic and tumor-suppressive roles depending on the cancer type. To investigate the role of miR-205 in mammary gland development in vivo, the authors generated a mouse model with miR-205 KO. They observed mammary gland hyperplasia with miR-205 KO, which is consistent with its tumor suppressing function.
While the mouse model itself would be helpful to the field in studying the function of miR-205 in the context of breast cancer, several other studies have introduced miR-205 KO mice (e.g. Wang et al 2013, Farmer et al 2017, Lu et al 2019). It would be more informative to the field if the authors could dive deeper into the physiological effect of miR-205 loss in the context of mammary gland biology and breast cancer using this model, and provide more mechanistic insights.
1. To generate miR-205 KO mice: How were the conditional allele introduced into the mice? What mouse crossing scheme was designed to generate miR-205 WT, het and KO mice?
2. Were littermate controls used in the experiment?
3. The method of Southern blot was not described clearly. Please elaborate.
4. The figures and supplementary figures should be introduced in the manuscript sequentially. Please introduce Supp Fig S2A before line 63 when Supp Fig S2B is mentioned.
5. Please make sure all oligonucleotide sequences involved in the experiment are included, such as the probe sequence for in situ hybridization and primer sequences for RT-PCR.
6. The authors confirmed the KO of miR-205 in different tissues by Southern and Nothern blot in Supp Fig S2. Since this study is primarily focused on miR-205 KO in mammary gland, why is mammary gland missing in Supp Fig S2?
7. The decimal points in Fig 2 and Supp Fig S3 were commas. Please correct.
8. The description of Supp Fig S4 in Line 72 did not match the content. Please correct.
9. Line 83-85 discussed the whole mount analysis of +/+ and -/- mammary glands at different stages without a figure reference. Please refer to the relevant figures.
10. Line 86-89 discussed the expression pattern of miR-205, which is not closely relevant to the mouse phenotypes of miR-205 KO discussed in adjacent paragraphs. Please edit to avoid disrupting the logic flow of the manuscript.
11. I did not see scale bars in Fig 2, please add those.
12. In the figure legend of Fig 2, the authors wrote that panel B was related to 5-month-old female mice, which did not match the labeling in the figures. Please correct.
13. Can the authors provide more background in mammary gland development? Why they chose to examine mice at 1.5 vs 5.5 months? And how did the differences in mammary gland morphology translate to its function?
14. Line 110-111 mentioned the consistency between their data and previous data on HER3 which is a direct target of miR-205. Have the authors examined the expression level of HER3 changed in the miR-205 KO mice?
15. Please elaborate the # of mice used in each experiment, especially Fig 2 when the authors compared morphological features of mammary gland between WT and KO. For RT-PCR, please also elaborate the # of replicates.
16. Please explain in more detail about how the branching and outgrowth were quantified and compared in Fig 2.
Comments on the Quality of English Language
The quality of English writing is good overall, with minor errors:
1. In Line 21-22, "Indeed, mammary glands of miR-205 -/- female mice at dif- 21 ferent ages (1.5 and 5.5 months) show increased outgrow", "outgrow" should be changed to "outgrowth".
2. At the beginning of Line 93, "indeed" and "unexpectedly" appeared at the same time, which is rarely seen. Please only retain one of them.
Author Response
REVIEWER 3.
- To generate miR-205 KO mice: How were the conditional allele introduced into the mice? What mouse crossing scheme was designed to generate miR-205 WT, het and KO mice? AND
- Were littermate controls used in the experiment? We better specified the protocol in the material and method section, and also specified that littermates were always used as controls.
- The method of Southern blot was not described clearly. Please elaborate. We better specified the protocol in the material and method section and in the Figure S1 legend.
- The figures and supplementary figures should be introduced in the manuscript sequentially. Please introduce Supp Fig S2A before line 63 when Supp Fig S2B is mentioned. We thank the reviewer for this comment. We actually modified the number of figures in the text, and made sure that in this revised version they are introduced in the manuscript sequentially.
- Please make sure all oligonucleotide sequences involved in the experiment are included, such as the probe sequence for in situ hybridization and primer sequences for RT-PCR. Both primers for in situ hybridization and qRT-PCR were commercially available, we specified now the information in the material and method section.
- The authors confirmed the KO of miR-205 in different tissues by Southern and Nothern blot in Supp Fig S2. Since this study is primarily focused on miR-205 KO in mammary gland, why is mammary gland missing in Supp Fig S2? We specifically assessed mIR-205 expression in the mammary gland by qRT-PCR and ISH assays.
- The decimal points in Fig 2 and Supp Fig S3 were commas. Please correct. We corrected (now Figures 1, 3 and 4).
- The description of Supp Fig S4 in Line 72 did not match the content. Please correct. Thank you for the note, we modified the text in the new version.
- Line 83-85 discussed the whole mount analysis of +/+ and -/- mammary glands at different stages without a figure reference. Please refer to the relevant figures. We modified the figures and the description.
- Line 86-89 discussed the expression pattern of miR-205, which is not closely relevant to the mouse phenotypes of miR-205 KO discussed in adjacent paragraphs. Please edit to avoid disrupting the logic flow of the manuscript. We thank the reviewer for this comment. We moved the description of miR-205 localization in the first paragraph.
- I did not see scale bars in Fig 2, please add those. We added scale bars in new Figure 3 and 4 of the revised version of the manuscript.
- In the figure legend of Fig 2, the authors wrote that panel B was related to 5-month-old female mice, which did not match the labeling in the figures. Please correct. We thank the reviewer for the note, we modified the figures and corrected the description.
- Can the authors provide more background in mammary gland development? Why they chose to examine mice at 1.5 vs 5.5 months? And how did the differences in mammary gland morphology translate to its function? We thank the reviewer for the suggestion to specify the reason of our choice: between 3 and 6 weeks of age, mice are in their puberal stage, when mammary gland outgrowth is more prominent, whereas after 8 weeks they can be considered mature virgin, and their gland is fully formed and characterized by multiple branches. 1.5 mo and 5.5 mo of ages have been thus been chosen as representative of puberal and mature virgin statuses, respectively, of mammary gland development. We better specified the choice rational in the text.
- Line 110-111 mentioned the consistency between their data and previous data on HER3 which is a direct target of miR-205. Have the authors examined the expression level of HER3 changed in the miR-205 KO mice? We did make an attempt to analyze HER3 expression by IHC in sections from miR-205 AND KO mice at both 1.5 and 5.5 months of age, unfortunately in our hands the staining was extremely faint, not evaluable. We did not have material left to assess protein expression by Western blot.
- Please elaborate the # of mice used in each experiment, especially Fig 2 when the authors compared morphological features of mammary gland between WT and KO. For RT-PCR, please also elaborate the # of replicates. We split Figure 2 in two figures (3 and 4), added the number of replicates, a plot and the statistics.
- Please explain in more detail about how the branching and outgrowth were quantified and compared in Fig 2. We better specified the method for branching and outgrowth quantification both in material and methods and in the legend of Figure 3.
MINOR COMMENTS: thank you, we corrected the spelling and grammar errors.

Round 2
Reviewer 2 Report
Comments and Suggestions for Authors
The authors have addressed the comments sufficiently.
Reviewer 3 Report
Comments and Suggestions for Authors
The revised version of manuscript is much improved in that: 1) The authors briefly reviewed previous miR-205 KO/KD models and their phenotypes, providing sufficient background for readers with diverse scientific background and thus the significance of the research is better delivered. 2) The clarity of both the main text and the method section were greatly improved.
The scale bars still seem missing from Figure 4A and Figure 5 in the revised manuscript. The authors should make sure that scale bars are always present with microscopic images such as H&E staining. Otherwise, in my opinion the manuscript is suitable for publication.